# Adaptively Aligned Image Captioning via Adaptive Attention Time

**Lun Huang**[1]    **Wenmin Wang**[1,3*]    **Yaxian Xia**[1]    **Jie Chen**[1,2]

[1]School of Electronic and Computer Engineering, Peking University
[2]Peng Cheng Laboratory
[3]Macau University of Science and Technology
huanglun@pku.edu.cn, {wangwm@ece.pku.edu.cn, wmwang@must.edu.mo}
xiayaxian@pku.edu.cn, chenj@pcl.ac.cn

## Abstract

Recent neural models for image captioning usually employ an encoder-decoder framework with an attention mechanism. However, the attention mechanism in such a framework aligns one single (attended) image feature vector to one caption word, assuming one-to-one mapping from source image regions and target caption words, which is never possible. In this paper, we propose a novel attention model, namely *Adaptive Attention Time* (AAT), to align the source and the target adaptively for image captioning. AAT allows the framework to learn how many attention steps to take to output a caption word at each decoding step. With AAT, an image region can be mapped to an arbitrary number of caption words while a caption word can also attend to an arbitrary number of image regions. AAT is deterministic and differentiable, and doesn't introduce any noise to the parameter gradients. In this paper, we empirically show that AAT improves over state-of-the-art methods on the task of image captioning. Code is available at https://github.com/husthuaan/AAT.

## 1   Introduction

Image captioning aims to automatically describe the content of an image using natural language [15, 31, 18, 7]. It is regarded as a kind of "translation" task that translates the content in an image to a sequence of words in some language.

Inspired by the development of neural machine translation [24], recent approaches to image captioning that adopt an encoder-decoder framework with an attention mechanism have achieved great success. In such a framework, a CNN-based image encoder is used to extract feature vectors for a given image, while an RNN-based caption decoder to generate caption words recurrently. At each decoding step, the attention mechanism attends to one (averaged) image region.

Despite the success that the existing approaches have achieved, there are still limitations in the current encoder-decoder framework. The attention model generates one single image feature vector at each decoding step, which subsequently provides the decoder information for predicting a caption word. Obviously, one-to-one mapping from image regions to caption words is assumed in the paradigm, which, however, is never possible and may involve the following issues: 1) unnecessary or even misleading visual information is given to all caption words, no matter whether they require visual clues or not [17]; 2) visual information in the decoder accumulates over time; however, words at earlier decoding steps need more knowledge about the image, but they are provided less [32]; 3) it's hard for the decoder to understand interactions among objects by analyzing one single attended

---

weighted averaged feature vector. To conclude, a decoder requires a different number of attention steps in different situations, which one-to-one mapping can't guarantee.

To this end, we develop a novel attention model, namely *Adaptive Attention Time* (AAT), to realize adaptive alignment from image regions to caption words. At each decoding step, depending on the decoder's confidence, AAT decides whether to take an extra attention step, or to output a caption word directly and move on to the next decoding step. If it's the former case, then at the subsequent attention step, AAT will take one more attended feature vector into the decoder. Again, AAT will face the same choice. The attention process proceeds, until the decoder is confident enough to output a word and move forward. With the techniques from *Adaptive Computation Time* (ACT) [9], we are able to make AAT deterministic and differentiable. Furthermore, we take advantage of the multi-head attention [25] to introduce fewer attention steps and make it easier for the decoder to comprehend interactions among objects in an image.

We evaluate the effectiveness of AAT by comparing it with a *base* attention model, which takes one attending step as one decoding step, and a *recurrent* attention model, which takes a fixed number of attention steps for each decoding step. We show that those two models are special cases of AAT, and AAT is superior to them with empirical results. Experiments also show that the proposed AAT outperforms previously published image captioning models. And one single image captioning model can achieve a new state-of-the-art performance of 128.6 CIDEr-D [26] score on MS COCO dataset offline test split.

## 2 Related Work

**Image Captioning.** Recently, neural-based encoder-decoder frameworks, which are inspired by the great development of neural machine translation [24], have replaced earlier rule/template-based [33, 23] approaches as the mainstream choice for image captioning. For instance, an end-to-end framework consisting of a CNN encoder and an LSTM decoder is proposed in [27]. Further in [29], the spatial attention mechanism on CNN feature map is introduced to let the deocder attend to different image regions at different decoding steps. More recently, semantic information such as objects, attributes and relationships are integrated to generate better descriptions [35, 2, 34, 30, 10]. However, all these methods assume simple and strict mappings from image regions to caption words, in the form of either one-to-all or one-to-one. To allow more mapping forms, efforts have been made in recent years: in [32], a *review* module is added to the encoder-decoder framework, which performs a number of *review* steps on the encoded image features between the encoding and the decoding processes and produces more representative *thought vectors*; in [17], an adaptive attention mechanism is proposed to decide when to activate the visual attention; in [11], a framework is designed with two agents which control the inputs and the outputs of the decoder respectively; [6] proposes to attend more times to the image per word.

**Adaptive Computation Time.** *Adaptive Computation Time* (ACT) is proposed as an algorithm to allow recurrent neural networks to learn how many computational steps to take between receiving an input and emitting an output. Later, ACT is utilized to design neural networks of a dynamic number of layers or operations [28, 5]. In this paper, we implement our *Adaptive Attention Time* (AAT) model by taking the advantage of ACT.

## 3 Method

Our image captioning model with *Adaptive Attention Time* (AAT) is developed upon the attention based encoder-decoder framework. In this section, we first describe the framework in Section 3.1, then show how we implement AAT to realize adaptive alignment in Section 3.2.

### 3.1 Model Framework

**Image Encoder.** The encoder in the attentive encoder-decoder framework is to extract a set of image feature vectors $A = \{\boldsymbol{a}_1, \boldsymbol{a}_2, \ldots, \boldsymbol{a}_k\}$ of different image regions for the given image, where $k$ is the number of image regions. A typical choice is a pre-trained Faster-RCNN [20] model [2].

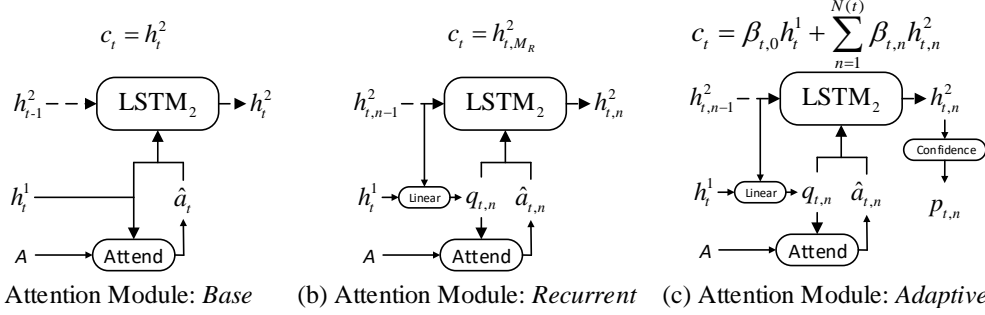

(a) Attention Module: *Base*  (b) Attention Module: *Recurrent*  (c) Attention Module: *Adaptive*

Figure 1: Different attention models. For each decoding step, **(a)** *base* takes one attention step; **(b)** *recurrent* takes fixed $M_R$ steps; **(c)** *adaptive* takes adaptive steps.

**Caption Decoder.** A two-layer LSTM decoder is popular in recent researches for image captioning [2]. We formalize a decoder with this kind of structure into three parts: an *input* module, an *attention* module and an *output* module.

The *input* module, which models the input word as well as the context information of the decoder, consists of an LSTM layer (LSTM$_1$). The process of this layer is:

$$(\boldsymbol{h}_t^1, \boldsymbol{m}_t^1) = \text{LSTM}_1\big([\Pi_t \boldsymbol{W}_e, \bar{\boldsymbol{a}} + \boldsymbol{c}_{t-1}], (\boldsymbol{h}_{t-1}^1, \boldsymbol{m}_{t-1}^1)\big) \tag{1}$$

where $\boldsymbol{h}_t^1, \boldsymbol{m}_t^1$ are the hidden state and memory cell of LSTM$_1$; $\boldsymbol{W}_e$ is a word embedding matrix; $\Pi_t$ is one-hot encoding of the input word at time step $t$; $\boldsymbol{c}_{t-1}$ is the previous context vector of the decoder, which is also the output of the *attention* module and input of the *output* module; and $\bar{\boldsymbol{a}} = \frac{1}{k}\sum_i \boldsymbol{a}_i$ is the mean-pooling of $A$ and is added to $\boldsymbol{c}_{t-1}$ to provide global information to the *input* module.

The *attention* module applies the proposed attention model on the image feature set $A$ and generates a context vector $\boldsymbol{c}_t$, which is named as *Adaptive Attention Time* (AAT) and will be described in the following section together with other comparing attention models.

The *output* module passes $\boldsymbol{c}_t$ through a linear layer with softmax activation to predict the probability distribution of the vocabulary:

$$p(y_t \mid y_{1:t-1}) = \text{softmax}(\boldsymbol{c}_t \boldsymbol{W}_p + \boldsymbol{b}_p) \tag{2}$$

## 3.2 Adaptive Alignment via Adaptive Attention Time

To implement adaptive alignment, we allow the decoder to take arbitrary attention steps for every decoding step. We first introduce the *base* attention mechanism which takes one attention step for one decoding step, then the *recurrent* version which allows the decoder to take a fixed number of attention steps, finally the *adaptive* version *Adaptive Attention Time* (AAT), which adaptively adjusts the attending time.

### 3.2.1 Base Attention Model

Common practice of applying attention mechanism to image captioning framework is to measure and normalize the attention score $\alpha_i$ of every candidate feature vector $\boldsymbol{a}_i$ with the given query (i.e. $\boldsymbol{h}_t^1$), resulting in one single weighted averaged vector $\hat{\boldsymbol{a}}_t = \sum_i^K \alpha_i \boldsymbol{a}_i$ over the whole feature vector set $A$.

By applying an attention mechanism to the image captioning framework, we obtain the attended feature vector:

$$\hat{\boldsymbol{a}}_t = \boldsymbol{f}_{att}(\boldsymbol{h}_t^1, A) \tag{3}$$

where $\boldsymbol{f}_{att}$ is the attention function. Next $\hat{\boldsymbol{a}}_t$ together with $\boldsymbol{h}_t^1$ is fed into another LSTM layer (LSTM$_2$):

$$(\boldsymbol{h}_t^2, \boldsymbol{m}_t^2) = \text{LSTM}_2\big([\hat{\boldsymbol{a}}_t, \boldsymbol{h}_t^1], (\boldsymbol{h}_{t-1}^2, \boldsymbol{m}_{t-1}^2)\big) \tag{4}$$

We let the context vector be the output hidden state: $\boldsymbol{c}_t = \boldsymbol{h}_t^2$.

### 3.2.2 Recurrent Attention Model

Rather than limiting the attention module to accessing the feature vectors only once for one decoding step. We allow attending for multiple times with *recurrent* attention.

First, we make the attention output vary with attention time, and we construct the attention query $q_{t,n}$ at attention time step $n$ of decoding time step $t$ by transforming $h_t^1$ and $h_{t,n-1}^2$ through a linear layer:

$$q_{t,n} = [h_t^1, h_{t,n-1}^2]W_q + b_q \tag{5}$$

where $h_{t,n-1}^2$ is the output of LSTM$_2$ at attention time step $n-1$ of decoding time step $t$ ($h_{t,0}^2 = h_{t-1}^2$).

Then $q_{t,n}$ is fed to the attention module to generate $\hat{a}_t = f_{att}(q_{t,n}, A)$, which is further fed to LSTM$_2$:

$$h_{t,n}^2, m_{t,n}^2 = \text{LSTM}_2\big([\hat{a_{t,n}}, q_{t,n}^1], (h_{t,n-1}^2, m_{t,n-1}^2)\big) \tag{6}$$

We set $h_t^2 = h_{t,M_r}^2, m_t^2 = m_{t,M_r}^2$, where $h_{t,M_r}^2, m_{t,M_r}^2$ are the hidden state and memory cell of the last attention step, and $M_r$ is the number of attention steps for each decoding step.

We let the context vector be the output hidden state at the last attention step: $c_t = h_{t,M_r}^2 = h_t^2$.

### 3.2.3 Adaptive Attention Time

We further allow the decoder attending to the image for arbitrary times with *Adaptive Attention Time*.

To determine how many times to perform attention, an extra confidence network is added to the output of LSTM$_2$, since it's used to predict probability distribution. We design the confidence network as a two-layer feed-forward network:

$$p_{t,n} = \begin{cases} \sigma\big(\max\big(0, h_t^1 W_1 + b_1\big) W_2 + b_2\big) & n = 0 \\ \sigma\big(\max\big(0, h_{t,n}^2 W_1 + b_1\big) W_2 + b_2\big) & n > 0 \end{cases} \tag{7}$$

where $\sigma$ denotes the sigmoid activation. The total required attention steps is determined by:

$$N(t) = \min\{n' : \prod_{n=0}^{n'}(1 - p_{t,n}) < \epsilon\} \tag{8}$$

where $\epsilon$ is a threshold, and is a small value slightly greater than 0 ($\epsilon = $ 1e-4 in this paper).

The final hidden state and memory cell of LSTM$_2$ are computed as:

$$\begin{cases} h_t^2 &= \beta_{t,0}h_t^1 + \sum_{n=1}^{N(t)}\beta_{t,n}h_{t,n}^2, \\ m_t^2 &= \beta_{t,0}m_{t-1}^2 + \sum_{n=1}^{N(t)}\beta_{t,n}m_{t,n}^2 \end{cases} \tag{9}$$

where

$$\beta_{t,n} = \begin{cases} p_{t,0} & n = 0 \\ p_{t,n}\prod_{n'=0}^{n-1}(1 - p_{t,n'}) & n > 0 \end{cases} \tag{10}$$

$h_t^1$ is added to show that we can obtain the context directly from the *input* module without attending to image feature vectors. And $c_{t-1}^2$ is to show that when deciding not to attend image features, the memory cell should maintain the previous state and not be updated.

**Normalization.** We normalize the hidden state and memory cell at each attention step: $h_{t,n}^2 \leftarrow$ LayerNorm$_h(h_{t,n}^2), m_{t,n}^2 \leftarrow$ LayerNorm$_m(m_{t,n}^2)$

We also normalize the weight of each attention step to make the sum of them to 1: $\beta_{t,n} \leftarrow \frac{\beta_{t,n}}{\sum_{n=0}^{N(t)}\beta_{t,n}}$.

**Time Cost Penalty.** We add a "attention time" loss to the training loss in order to penalize the time cost for the attention time steps:

$$L_t^a = \lambda\big(N(t) + \sum_{n=0}^{N(t)}(n+1)(1 - p_{t,n})\big) \tag{11}$$

where $\lambda$ is a hyper-parameter; $(n+1)(1-p_{t,n})$ encourages $p_{t,n}$ to be larger so that the total attention steps can be reduced; and $N(t)$ is added to indicate the attention time steps and doesn't contribute to the parameter gradients.

**Minimum and Maximum Attention Steps.** We can set a minimum attention step $M_{min}$ to make the attention module takes at least $M_{min}$ attention steps for each decoding step, by simply setting $p_{t,n} = 0$ for $0 <= n <= M_{min} - 1$.

We can also set a maximum attention steps $M_{max}$ to make sure the attention module takes at most $M_{max}$ attention steps by modifying $N(t)$:

$$N(t) = \min\{M_{max}, \min\{n' : \prod_{n=0}^{n'}(1-p_{t,n}) < \epsilon\}\} \tag{12}$$

As can be seen, the process of AAT is deterministic and can be optimized directly.

We let the context vector be the weighted average over all hidden states at all attention steps: $\boldsymbol{c}_t = \beta_{t,0}\boldsymbol{h}_t^1 + \sum_{n=1}^{N(t)} \beta_{t,n}\boldsymbol{h}_{t,n}^2 = \boldsymbol{h}_t^2$.

### 3.2.4 Connections between Different Attention Models

*Base* attention model is a special case of *recurrent* attention model when $M_r = 1$, and *recurrent* attention model is a special case of *adaptive* attention model when $M_{max} = M_{min} = M_r$.

## 4 Experiments

### 4.1 Dataset, Settings, and Metrics

**Dataset.** We evaluate our proposed method on the popular MS COCO dataset [16]. MS COCO dataset contains 123,287 images labeled with at least 5 captions, including 82,783 for training and 40,504 for validation. MS COCO also provides 40,775 images as the test set for online evaluation. We use the "Karpathy" data split [13] for the performance comparisons, where 5,000 images are used for validation, 5,000 images for testing, and the rest for training.

**Settings.** We convert all sentences to lower case, drop the words that occur less than 5 times, and trim each caption to a maximum of 16 words, which results in a vocabulary of 10,369 words.

Identical to [2], we employ Faster-RCNN [20] pre-trained on ImageNet and Visual Genome to extract bottom-up feature vectors of images. The dimension of the original vectors is 2048, and we project them to the dimension of $d = 1024$, which is also the hidden size of the LSTM of the decoder and the size of word embedding.

As for the training process, we train our model under cross-entropy loss for 20 epochs with a mini-batch size of 10, and ADAM [14] optimizer is used with a learning rate initialized with 1e-4 and annealed by 0.8 every 2 epochs. We increase the probability of feeding back a sample of the word posterior by 0.05 every 3 epochs [4]. Then we use self-critical sequence training (SCST) [21] to optimize the CIDEr-D score with REINFORCE for another 20 epochs with an initial learning rate of 1e-5 and annealed by 0.5 when the CIDEr-D score on the validation split has not improved for some training steps.

**Metrics.** We use different metrics, including BLEU [19], METEOR [22], ROUGE [8], CIDEr-D [26] and SPICE [1], to evaluate the proposed method and compare with others. All the metrics are computed with the publicly released code[2].

### 4.2 Ablative Studies

**Attention Time Steps (Attention Model).** To show the effectiveness of adaptive alignment for image captioning, we compare the results of different attention models with different attention time steps, including *base*, which uses the conventional attentive encoder-decoder framework and forces

Table 1: Ablative studies of attention time steps. We show the results of different attention models with different attention time steps, which are reported after the cross-entropy loss training stage and the self-critical loss training stage. We obtain the mean score and the standard deviation of a metric for a model by training it for 3 times, each with a different seed for random parameter initialization.

| Model | Attention Time Steps | | | Cross-Entropy Loss | | | Self-Critical Loss | | |
|---|---|---|---|---|---|---|---|---|---|
| | min. | max. | avg. | METEOR | CIDEr-D | SPICE | METEOR | CIDEr-D | SPICE |
| Base | 1 | 1 | 1 | $27.58 \pm 0.03$ | $113.37 \pm 0.15$ | $20.76 \pm 0.03$ | $27.96 \pm 0.02$ | $123.76 \pm 0.41$ | $21.35 \pm 0.10$ |
| Recurrent | 2 | 2 | 2 | $27.66 \pm 0.06$ | $114.21 \pm 0.35$ | $20.84 \pm 0.05$ | $28.06 \pm 0.05$ | $124.22 \pm 0.24$ | $21.54 \pm 0.10$ |
| | 4 | 4 | 4 | $27.79 \pm 0.01$ | $114.72 \pm 0.17$ | $20.93 \pm 0.08$ | $28.14 \pm 0.05$ | $124.96 \pm 0.33$ | $21.75 \pm 0.03$ |
| | 8 | 8 | 8 | $27.75 \pm 0.05$ | $114.74 \pm 0.23$ | $20.93 \pm 0.05$ | $28.16 \pm 0.02$ | $124.85 \pm 0.30$ | $21.76 \pm 0.01$ |
| Adaptive | 0 | 4 | $2.55 \pm 0.01$ | $\mathbf{27.82 \pm 0.02}$ | $\mathbf{115.12 \pm 0.38}$ | $\mathbf{20.94 \pm 0.07}$ | $\mathbf{28.25 \pm 0.05}$ | $\mathbf{126.48 \pm 0.22}$ | $\mathbf{21.84 \pm 0.05}$ |

Table 2: Tradeoff for time cost penalty. We show the average attention time steps as well as the performance with different values of $\lambda$. The results are reported by a single model after self-critical training stage.

| $\lambda$ | avg. steps | BLEU-1 | BLEU-2 | BLEU-3 | BLEU-4 | ROUGE | CIDEr-D | METEOR | SPICE |
|---|---|---|---|---|---|---|---|---|---|
| 1e-1 | 0.35 | 78.1 | 62.1 | 47.4 | 35.8 | 56.9 | 118.7 | 27.2 | 20.5 |
| 1e-3 | 1.03 | 80.0 | 64.3 | 49.7 | 37.8 | 58.0 | 125.7 | 28.2 | 21.6 |
| 1e-4 | 2.54 | **80.1** | **64.6** | **50.1** | **38.2** | **58.2** | **126.7** | **28.3** | **21.9** |
| 1e-5 | 3.77 | 80.0 | **64.6** | 50.0 | **38.2** | **58.2** | 126.5 | **28.3** | 21.8 |
| 0 | 4.00 | 80.0 | 64.5 | **50.1** | **38.2** | **58.2** | **126.7** | **28.3** | 21.8 |

to align one (attended) image region to one caption word; *recurrent*, which at each decoding step attends to a fixed number of image regions recurrently; and *adaptive*, the proposed method in this paper, which attends to an adaptive number of image regions at each decoding step, allowing adaptive alignment from image regions to caption words.

From Table 1, we observe that: 1) *Recurrent* attention model (slightly) improves *base* attention model for all metrics; 2) Enlarging the attention time steps for *recurrent* attention improves the performance; 3) *Adaptive* attention model (AAT) further outperforms *recurrent* attention while requiring an average attention time steps of 2.55, which is smaller comparing to 4 and 8 attention time steps of the *recurrent* attention model. It shows that: 1) incorporating more attention steps by adopting *recurrent* attention model helps to obtain better performance, but increases the computation cost linearly with the number of the recurrent steps; 2) adaptively aligning image feature vectors to caption words via *Adaptive Attention Time* requires less computation cost meanwhile leading to further better performance. This demonstrates the superiority of AAT and indicates it to be a general solution to such sequence to sequence learning tasks as image captioning.

**Tradeoff for Time Cost Penalty.** We show the effect of $\lambda$, the penalty factor for attention time cost, in Table 2. We assign different values to $\lambda$ and obtain the corresponding results of average attention time as well as performance after the cross-entropy loss training stage. From Table 2, we observe that: 1) smaller value of $\lambda$ leads to more attention time and relatively higher performance; 2) the increment of performance gain will stop at some value of $\lambda$ but the attention time won't as $\lambda$ decreases. We find: for $\lambda = 1e-4$, it has the best performance while involves relatively small attention time steps; for $\lambda = 1e-3$, it requires only 1.03 averaged attention steps but outperforms both the *base* and the *recurrent* models.

**Number of Attention Heads.** There are two kinds of commonly used attention functions: additive attention [3] and dot-product attention. The former is popular among image captioning models while the latter has also shown its success by employing multiple attention heads [25]. We compare the two attention functions and explore the impact of using different numbers of attention heads. From Table 3, we find that: 1) when using one single head, additive attention has better performance than dot-product attention with higher scores for all metrics and fewer attention steps; 2) as the number of attention heads increases, the number of attention steps reduces and the performance first goes up then down; 3) 4 and 8 attention heads of dot-product attention bring better performance than other alternatives.

Table 3: Ablation study of attention type and number of attention heads. For all models, we set $\lambda$ =1e-4. The results are reported by a single model after self-critical training stage.

| Type | Head(s) | avg. steps | BLEU-1 | BLEU-2 | BLEU-3 | BLEU-4 | ROUGE | CIDEr-D | METEOR | SPICE |
|------|---------|-----------|--------|--------|--------|--------|-------|---------|--------|-------|
| Addictive | 1 | 2.54 | 80.1 | 64.6 | 50.1 | 38.2 | 58.2 | 126.7 | 28.3 | 21.9 |
| Dot-product | 1 | 2.84 | 79.9 | 64.5 | 50.0 | 38.1 | 58.0 | 126.3 | 28.2 | 21.8 |
| | 2 | 2.78 | 80.0 | 64.6 | 50.1 | 38.3 | 58.2 | 126.8 | 28.2 | 21.9 |
| | 4 | 2.75 | **80.3** | **65.0** | **50.5** | 38.6 | 58.4 | 127.7 | 28.4 | 22.0 |
| | 8 | 2.66 | 80.1 | 64.9 | **50.5** | **38.7** | **58.5** | **128.6** | **28.6** | **22.2** |
| | 16 | 2.64 | 80.1 | 64.7 | 50.2 | 38.4 | 58.3 | 128.0 | 28.4 | 22.1 |

Table 4: Single model performance of other state-of-the-art methods as well as ours on the MS-COCO "Karpathy" test split. For our AAT model, we use multi-head attention [25] with the number of attention heads to be 8 and $\lambda$ =1e-4.

| Method | Cross-Entropy Loss | | | | | Self-Critical Loss | | | | |
|--------|--------|--------|-------|---------|-------|--------|--------|-------|---------|-------|
| | BLEU-4 | METEOR | ROUGE | CIDEr-D | SPICE | BLEU-4 | METEOR | ROUGE | CIDEr-D | SPICE |
| LSTM [27] | 29.6 | 25.2 | 52.6 | 94.0 | - | 31.9 | 25.5 | 54.3 | 106.3 | - |
| ADP-ATT [17] | 33.2 | 26.6 | - | 108.5 | - | - | - | - | - | - |
| SCST [21] | 30.0 | 25.9 | 53.4 | 99.4 | - | 34.2 | 26.7 | 55.7 | 114.0 | - |
| Up-Down [2] | 36.2 | 27.0 | 56.4 | 113.5 | 20.3 | 36.3 | 27.7 | 56.9 | 120.1 | 21.4 |
| RFNet [12] | 35.8 | 27.4 | 56.8 | 112.5 | 20.5 | 36.5 | 27.7 | 57.3 | 121.9 | 21.2 |
| GCN-LSTM [34] | 36.8 | 27.9 | 57.0 | 116.3 | 20.9 | 38.2 | 28.5 | 58.3 | 127.6 | 22.0 |
| SGAE [30] | - | - | - | - | - | 38.4 | 28.4 | **58.6** | 127.8 | 22.1 |
| AAT (Ours) | **37.0** | **28.1** | **57.3** | **117.2** | **21.2** | **38.7** | **28.6** | 58.5 | **128.6** | **22.2** |

## 4.3 Comparisons with State-of-the-arts

**Comparing Methods.** The compared methods include: LSTM [27], which encodes the image using a CNN and decode it using an LSTM; ADP-ATT [17], which develops a visual sentinel to decide how much to attend to images; SCST [21], which employs a modified visual attention and is the first to use Self Critical Sequence Training(SCST) to directly optimize the evaluation metrics; Up-Down [2], which employs a two-layer LSTM model with bottom-up features extracted from Faster-RCNN; RFNet [12], which fused encoded results from multiple CNN networks; GCN-LSTM [34], which predicts visual relationships between any two entities in the image and encode the relationship information into feature vectors; and SGAE [30], which introduces language inductive bias into its model and applies auto-encoding scene graphs.

**Analysis.** We show the performance of one single model of these methods under both the cross-entropy loss training stage and the self-critical loss training stage. It can be seen that: for the cross-entropy stage, AAT achieves the highest scores among all compared methods for all metrics (BLEU-4, METEOR, ROUGE, CIDEr-D, and SPICE); and for the self-critical stage, AAT reports the highest scores for most metrics, with the ROUGE score slightly lower than the highest (58.5 vs. 58.6).

Comparing to Up-Down [2], which is the previous state-of-the-art model and is the most similar to our AAT model in terms of the model framework, our single AAT model improves the scores on BLEU-4, METEOR, ROUGE, CIDEr-D, and SPICE by 2.2%, 4.1%, 1.6%, 3.3%, and 4.4% respectively for cross-entropy loss training stage by 6.6%, 3.2%, 2.8%, 7.1%, and 3.7% respectively for self-critical loss training stage, which indicates a significant margin and shows that AAT is superior to the vanilla attention model.

## 4.4 Qualitative Analysis

To gain a qualitative understanding of our proposed method, we use two examples and visualize the caption generation process of a model that employs AAT and a 4-head attention in Figure 2. For each example, we show the attention steps taken at each decoding step, with the visualized attention regions for all the 4 attention heads, the confidence for the output at the current attention step, and the corresponding weight. We also show the confidence/weight for the non-visual step (colored in orange and before the attention steps), which corresponds to $h_t^1$ of the *input* module in the decoder and contains the previous decoding information and is used to avoid attention steps.

We observe that: 1) the attention regions of each attention heads are different from others, which indicates that every attention head has its own concern; 2) the confidence increases with the attention steps, which is like human attention: more observing leads to better comprehension and higher confidence; 3) the number of attention steps required at different decoding steps is different, more steps are taken at the beginning of the caption or a phase, such as "on the side" and "at a ball", which indicates that adaptive alignment is effective and has been realized by AAT.

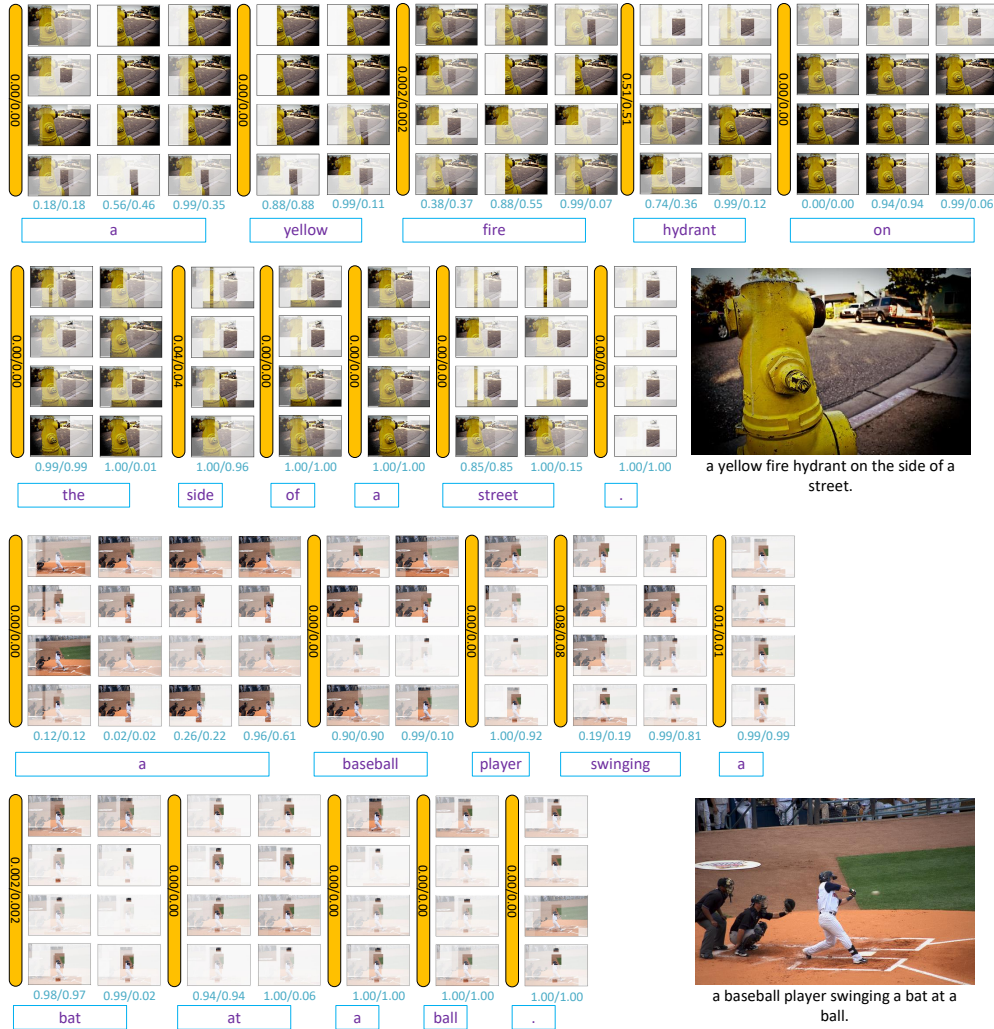

Figure 2: Qualitative examples for the caption generation process of AAT. We show the attention steps taken at each decoding step, with the visualized attention regions, the confidence and the weights of each attention step (*confidence/weight* is shown below the attention regions for each step).

## 5 Conclusion

In this paper, we propose a novel attention model, namely Adaptive Attention Time (**AAT**), which can adaptively align image regions to caption words for image captioning. AAT allows the framework to learn how many attention steps to take to output a caption word at each decoding step. AAT is also generic and can be employed by any sequence-to-sequence learning task. On the task of image captioning, we empirically show that AAT improves over state-of-the-art methods. In the future, it will be interesting to apply our model to more tasks in computer vision such as video captioning and those in natural language processing such as machine translation and text summarization, as well as any model that can be modeled under the encoder-decoder framework with an attention mechanism.

## Acknowledgment

This project was supported by National Engineering Laboratory for Video Technology - Shenzhen Division, National Natural Science Foundation of China (NSFC, 61872256, 61972217), and The Science and Technology Development Fund of Macau (FDCT, 0016/2019/A1). We would also like to thank the anonymous reviewers for their insightful comments.

## Footnotes

[2]https://github.com/tylin/coco-caption

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
