[Reviews · NeurIPS 2019]

Reviewer 1



Originality: The work is moderately novel — the idea of ACT has been modified to propose a recurrent attention ahead. Quality: The paper is technically sound and is grounded in previous work. Clarity: The paper is well-written and easy to follow. Significance: While the approach itself is reasonably straight-forward, the empirical analysis and experimental results show the merit of using adaptive attention, that maybe of interest to the sequence modeling community in general.

Reviewer 2



- The paper combines the idea of Adaptive Computation Time (ACT) and multi-head attention build an attention mechanism called Adaptive Attention Time (AAT). Although the two techniques have been well explored individually, this is the first work combining it for attention for image captioning. - The paper is also clearly written with explanation of all the hyper-parameters used in the paper. This should make reproducing the results easier. - It is not clear what is the contribution of AAT compared to multi-head attention. The base attention model already is doing much better than up-down attention and recent methods like GCN-LSTM and so it’s not clear where the gains are coming from. It’d be good to see AAT applied to traditional single-head attention instead of multi-head attention to convincingly show that AAT helps. - More analysis is required to find the reason for improvement from recurrent attention model to adaptive attention model. For instance, how does the attention time steps vary with word position in the caption? Does this number change significantly after self-critical training? - How much does the attention change over multiple attention steps for each word position? From the qualitative results in the supplementary, it’s not clear, how is the attention changing from one attention step to another. - Is it the case that self-critical training is necessary to fully utilize the potential of AAT. The gains when trained just using Cross-Entropy Loss are minimal. Even for self-critical training, the gains in other metrics (SPICE, METEOR ) are minimal. Minor Comments: - In Line 32, the paper says that words at early decoding steps have little access to image information. Why is this the case for traditional models? Doesn’t every time step have access to the same set of visual features? - Are the ablations in Table 1 done on the same split as Table 2? Update [Post Rebuttal]: The authors addressed several of my concerns through experiments and answered many of my questions in the rebuttal. They showed that AAT helps even in case of single-head attention; that self-critical training is required to fully optimize the potential of AAT; Fixing attention steps introduces redundant or even misleading information since not all words require visual clues and that increasing the min number of steps reduces performance supporting the claim that adaptive attention time works better than recurrent attention. In light of the rebuttal, I am increasing my rating to 7.

Reviewer 3



Authors properly justify that for image captioning several attention steps (in the decoder) is reasonable. Also fixing the number of attention steps as in recurrent attention modules does not yield the best results. Their target task is image captioning. The model architecture that they use for encoding image is a standard Faster-RCNN pre-trained on ImageNet and Visual Genome. For the decoder they use a attention-based LSTM model. They augment the attention module by outputting a confidence score (through an MLP on hidden state) at each step and halting the recurrent attention as soon as the confidence score drops bellow threshold. They use similar loss as ACT (graves, 2016) to encourage model toward fewer steps. In the end by allowing their model to take between 0-4 attention steps, they have an average 2.2 steps, while getting better performance in compare to a recurrent attention baseline with 2, 4 or 8 steps (fixed). Their ablation study is helpful as it clarifies the effect of the loss (scst vs ce), number of attention steps, and the lambda factor for the act loss. *********************************************************************** Thank you for answering the comments. I still believe this is a grounded and strong work and I will keep my score at 7.

[Author Response · NeurIPS 2019]

**R1: Cut down on some sections (3.2.1, 3.2.2 and 3.2.5) to spare space for the qualitative examples.**

We will revise our paper according to the suggestion in the final version.

Table 1: Experiments on MS-COCO and Flicker30k datasets using single-head attention. (Row *Steps* shows the min./max./avg. attention time steps of each model.)

| Model | MS-COCO | | | | | | | Flicker30k | | | | | | |
|---|---|---|---|---|---|---|---|---|---|---|---|---|---|---|
| | Steps | Cross-Entropy Loss | | | Self-Critical Loss | | | Steps | Cross-Entropy Loss | | | Self-Critical Loss | | |
| | | M | C | S | M | C | S | | M | C | S | M | C | S |
| Base | 1/1/1 | 27.8 | 115.1 | 20.9 | 28.3 | 122.9 | 21.9 | 1/1/1 | 22.3±0.04 | 60.6±0.3 | 16.8±0.05 | **22.5±0.07** | 68.2±0.2 | 16.4±0.03 |
| Recurrent | 2/2/2 | **28.0** | 116.1 | **21.1** | 28.4 | 124.0 | 21.9 | 2/2/2 | **22.4±0.04** | 60.8±0.1 | 16.7±0.1 | **22.5±0.1** | 68.7±0.3 | **16.7±0.05** |
| | 4/4/4 | 27.8 | 115.1 | 20.9 | 28.4 | 124.2 | 21.9 | 4/4/4 | **22.4±0.03** | 61.3±0.5 | 16.7±0.04 | **22.5±0.06** | 68.8±0.4 | 16.5±0.05 |
| Adaptive | **0/4/2.4** | **28.0** | **116.5** | **21.1** | **28.5** | **126.8** | **22.0** | 0/4/2.3 | **22.4±0.03** | **61.5±0.4** | **16.9±0.03** | **22.5±0.03** | **69.2±0.3** | **16.7±0.03** |
| | 1/4/2.8 | 27.9 | 115.4 | **21.1** | 28.3 | 123.5 | **22.0** | | | | | | | |
| | 2/4/3.2 | 27.8 | 114.7 | **21.1** | 28.3 | 123.6 | **22.0** | | | | | | | |

**R2: Apply AAT on traditional single-head instead of multi-head attention to show that AAT helps.**

We added experiments on MS-COCO and Flicker30k using single-head attention, Table 1. As can be seen, *adaptive*
attention model with (0/4/2.4) yields best results, which show that AAT also helps single-head attention.

**R2: The base attention model performs better than up-down and GCN-LSTM.**

The reason lies in that the *base* attention model adopts a different structure ($LSTM_1$ in Section 3.1) and different
experimental settings (batch size, learning rate and schedule sampling rate in Section 4.1).

**R2: Provide more analysis to find the reason for improvement from *recurrent* attention model to *adaptive* atten-**
**tion model.**

The reason for *adaptive* attention model (AAT) improves from *recurrent* is that AAT helps to decide how many attention
steps (from zero to multiple, adaptively) to take before outputting a word, while the number of attention steps is fixed
for *recurrent*. Fixing attention steps introduces redundant or even misleading information since not all words require
visual clues [14]. In addition, our experimental results showed that increasing the number of *min.* attention steps for
*adaptive* attention model (1/4/2.8 and 2/4/3.2) degrades the performances, in Table 1.

**R2: How much does the attention change over multiple attention steps for each word position?**

It changed very much as shown in Fig. 1 in the appendix. For each word, the attention changes: **a)** towards more
accurate objects than previous steps; **b)** for objects which have connections with each other to obtain a better overview.

**R2: How does the attention time steps vary with word position?**

The numbers of attention time steps at the beginning of the sentence or phrases (*e.g* 'on the side" and "at a ball") are
larger than those at other positions.

**R2: Does this number change significantly after self-critical training?**

It doesn't change significantly after self-critical training but requires relatively less attention steps.

**R2: Is it the case that self-critical training is necessary to fully utilize the potential of AAT?**

Yes. We experimentally found that self-critical training significantly boosted the performance (Table 1 in the paper).

**R2: Why words at early decoding steps have little access to image information?**

Because the decoder incoorperates little information about the image at early steps.

**R2: Are the ablations in Table 1 done on the same split as Table 2 (in the main paper)?**

Yes, all the experiments in this paper are done on the 'Karpathy' splits.

**R3: Add flicker results and report STD.**

We experimented on Flicker-30k and reported results as well as STD in Table 1. STD on COCO dataset will be added
to the final version.

**R3: N(t) in eq. 14 is non-differentiable.**

$N(t)$ doesn't contribute for the gradients, and it solely indicates the number of attention steps.

[Meta-Review · NeurIPS 2019]

After feedback and reviewer discussion, this paper received final ratings of 6, 7 and 7. Although the novelty of the proposed model is relatively minor in the context of previous work proposing Adaptive Computation Time (Graves 2016), the reviewers were impressed by the empirical performance and praised the detailed ablation studies (including the additional experiments with single-headed attention in the author feedback, which was important in reaching the final consensus view of reviewers to accept this paper). We encourage the authors to follow the suggestion of R1 (cut down space devoted to standard captioning components in Secs 3.2.1, 3.2.2 and 3.2.5) in order to make space in the final version for the experiments from the author feedback.